# Cold Adaptation in Antarctic Notothenioids: Comparative Transcriptomics Reveals Novel Insights in the Peculiar Role of Gills and Highlights Signatures of Cobalamin Deficiency

**DOI:** 10.3390/ijms22041812

**Published:** 2021-02-11

**Authors:** Federico Ansaloni, Marco Gerdol, Valentina Torboli, Nicola Reinaldo Fornaini, Samuele Greco, Piero Giulio Giulianini, Maria Rosaria Coscia, Andrea Miccoli, Gianfranco Santovito, Francesco Buonocore, Giuseppe Scapigliati, Alberto Pallavicini

**Affiliations:** 1Department of Life Sciences, University of Trieste, 34127 Trieste, Italy; fansalon@sissa.it (F.A.); torboli_valentina@hotmail.com (V.T.); nickforna@gmail.com (N.R.F.); SAMUELE.GRECO@phd.units.it (S.G.); giuliani@units.it (P.G.G.); pallavic@units.it (A.P.); 2International School for Advanced Studies, 34136 Trieste, Italy; 3Department of Cell Biology, Charles University, 12800 Prague, Czech Republic; 4Institute of Biochemistry and Cell Biology, National Research Council of Italy, 80131 Naples, Italy; mr.coscia@ibp.cnr.it; 5Department for Innovation in Biological, Agro-Food and Forest Systems, University of Tuscia, 01100 Viterbo, Italy; andrea.miccoli@unitus.it (A.M.); fbuono@unitus.it (F.B.); scapigg@unitus.it (G.S.); 6Department of Biology, University of Padua, 35131 Padua, Italy; gianfranco.santovito@unipd.it; 7Anton Dohrn Zoological Station, 80122 Naples, Italy; 8National Institute of Oceanography and Experimental Geophysics, 34010 Trieste, Italy

**Keywords:** Cryonotothenioidea, cold adaptation, transcobalamin, Antarctica, RNA-seq

## Abstract

Far from being devoid of life, Antarctic waters are home to Cryonotothenioidea, which represent one of the fascinating cases of evolutionary adaptation to extreme environmental conditions in vertebrates. Thanks to a series of unique morphological and physiological peculiarities, which include the paradigmatic case of loss of hemoglobin in the family Channichthyidae, these fish survive and thrive at sub-zero temperatures. While some of the distinctive features of such adaptations have been known for decades, our knowledge of their genetic and molecular bases is still limited. We generated a reference de novo assembly of the icefish *Chionodraco hamatus* transcriptome and used this resource for a large-scale comparative analysis among five red-blooded Cryonotothenioidea, the sub-Antarctic notothenioid *Eleginops maclovinus* and seven temperate teleost species. Our investigation targeted the gills, a tissue of primary importance for gaseous exchange, osmoregulation, ammonia excretion, and its role in fish immunity. One hundred and twenty genes were identified as significantly up-regulated in Antarctic species and surprisingly shared by red- and white-blooded notothenioids, unveiling several previously unreported molecular players that might have contributed to the evolutionary success of Cryonotothenioidea in Antarctica. In particular, we detected cobalamin deficiency signatures and discussed the possible biological implications of this condition concerning hematological alterations and the heavy parasitic loads typically observed in all Cryonotothenioidea.

## 1. Introduction

The Drake Passage opening and the Tasman Gateway (20–30 Mya) isolated the Antarctic region from South America and Australia [1]. These changes were responsible for developing the Antarctic Circumpolar Current (ACC), which created a barrier that isolated the Southern Ocean, preventing the mixing of its waters with the Indian, Pacific, and Atlantic Oceans and triggering the cooling of Antarctica [2,3,4]. Within this context, only organisms possessing an appropriate physiological background could successfully adapt to freezing temperatures, while other species went extinct or migrated northwards to more suitable environments. These events led to the genetic and morphological differentiation between the Antarctic (i.e., Cryonotothenioidea) and non-Antarctic Notothenioidei within the order Perciformes [5,6]. The successful colonization of the hostile yet extremely stable Antarctic environment by fishes was made possible thanks to the development of several characteristic physiological, morphological, and molecular adaptations. These peculiar features include the production of anti-freeze glycoproteins (AFGPs), the loss of inducible heat shock response (HSR), the optimization of metabolic processes at cold temperatures, the lack of ossified bones, and the presence of high tissue lipid contents to compensate for the loss of the swim bladder [7,8,9]. Most notably, the blood of Cryonotothenioidea shows a reduced mean hemoglobin content, which culminates in the loss of this oxygen transporter in some species, coupled with remarkable modifications of their cardio-circulatory system [10].

In a standard marine environment, the absence of red blood cells and hemoglobin would not be compatible with life. However, icefishes are a remarkable evolutionary example of the interaction between environment, genetics, and selective pressure. The survival of these organisms in the Antarctic region is possible thanks to the low average temperature of seawater (−2 °C), which contributes to increased solubility of gases in fluids, potentially allowing their direct transport by plasma without the intervention of carrier molecules [11,12]. Moreover, icefish have evolved an implemented cardiovascular system that optimizes the transport of gases, including increased blood volume (2–5 times higher than red-blooded teleosts), a larger heart, and blood vessels with a larger diameter compared to their temperate relatives [13,14,15].

From an evolutionary and taxonomical perspective, Cryonotothenioidea include five out of the eight families recognized within Notothenioidea according to the World Register of Marine Species, i.e., Harpagiferidae, Artedidraconidae, Bathydraconidae, Channichthyidae, and Nototheniidae [6], whereas non-Antarctic notothenioids include Bovichtidae, Eleginopsidae, and Pseudaphritidae. However, it is worth noting that recent molecular investigations have challenged the monophyly of some of these clades [16]. Among these, Channichthyidae, nicknamed “icefish” or “white-blooded” fish, display the most distinctive morphological and physiological features, being the only vertebrates to lack hemoglobin and functional erythrocytes in adult life stages. While it is currently a matter of debate whether hemoglobin loss is an adaptive trait, or rather the product of genetic drift [10], the underlying cause was confirmed to reside in a gene loss event [12,17]. Despite the presence of red blood and normal hematocrit levels, the four other Cryonotothenioidea families show lower mean cellular hemoglobin concentrations than their non-Antarctic relatives and can survive carbon monoxide poisoning. This condition would be otherwise lethal in any other fish species [18].

Our understanding of these fascinating adaptation mechanisms is still incomplete, and the molecular machinery underpinning these unique developmental programs is mostly unknown. However, genetic and molecular information for Channichthyidae and red-blooded Cryonotothenioidei is becoming increasingly accessible. Following the release of the first fully-sequenced genome of a red-blooded species [19], other studies have targeted the blackfin icefish *Chaenocephalus aceratus* [17], the dragonfish *Parachaenichthys charcoti* [20], and the icefish *Chionodraco myersi* [21], revealing the presence of lineage-specific gene family expansions and the critical importance of mitochondrial function in cold adaptation. Most recently, large-scale genome sequencing approaches, carried out under the Vertebrate Genome Project (VGP) umbrella, led to the release of eleven draft genome assemblies for Notothenioidei (as of 20 January 2021).

Transcriptome studies can provide important complementary information to genome sequencing approaches, enabling to extend the range of functional investigations to the layer of gene expression [21]. Several RNA-sequencing studies have been carried out over the past few years in red- [22,23] and white-blooded Cryonotothenioidea [24,25,26]. However, just a handful experiments have targeted the gills, a highly vascularized tissue of fundamental importance for gaseous exchange [27,28], ammonia excretion [29], osmoregulation [30], and immunity [31], and none has ever targeted the gills of a hemoglobin-less species.

In the present work, we undertook a large-scale comparative study between the gill transcriptome of Antarctic and non-Antarctic Eupercaria, with a particular emphasis on Cryonotothenioidea, using the benthic icefish *Chionodraco hamatus* (Lönnberg, 1905) as a model species (Figure 1). These analyses were carried out to detect novel molecular adaptations to cold and to provide complementary functional information to those recently revealed by whole-genome sequencing studies [17,19,21]. Besides several previously described adaptations, we unexpectedly detected signatures of cobalamin deficiency, and we discuss their possible intimate link of our findings with the high level of parasitism and the hematological alterations found in all Cryonotothenioidea.

## 2. Results and Discussion

### 2.1. De Novo Assembly of the C. hamatus Gills Transcriptome

The reference transcriptome obtained from the gills of *C. hamatus* comprised 28,644 contigs with an average length of 1705 nucleotides and a N50 value of 2724 (Figure 1). The number of protein-coding genes annotated in the fully sequenced genomes of other Antarctic notothenioids is similar to that of other teleosts, i.e., 38,127 in *C. myersi* [21], 32,112 in *Trematomus bernacchii* (unpublished), 30,850 in *Gymnodraco acuticeps* (unpublished), 30,773 in *C. aceratus* [17], 30,381 in *P. charcoti* [20], and 26,850 in *N. coriiceps* [19]. As expected from the single-tissue origin of RNA-seq data, the *C. hamatus* transcriptome was slightly less complete than these genomic resources, with 66% present actinopterygian Benchmarking Universal Single-Copy Orthologs (BUSCOs) and just 11% and 23% fragmented and missing orthologs, respectively. These metrics were in line with other nototheniod transcriptomes obtained from single tissues [23], but lower than those obtained from most published notothenioid genomes, likely due to the lack of expression of several tissue-specific or developmentally regulated genes in the gills.

In further support to the quality of the reference gills transcriptome, 88.59% of the total reads could be mapped in pairs to assembled contigs and 13.65% as broken pairs, possibly due to transcript fragmentation. The low ambiguous mapping rate (0.04%) was in line with the near-complete lack of duplicated BUSCOs, highlighting that the reference transcriptome included non-redundant sequences. Annotation rates were also satisfying, considering the presence of non-coding transcripts and lineage-specific protein-coding genes with no similarity to proteins deposited in public databases. Overall, 15,362 contigs (53.63% of the total) could be annotated either with GO terms or conserved protein domain annotations (Figure 1).

Although *C. hamatus* was previously targeted by RNA-sequencing approaches to elucidate alterations of the hematopoietic pathways and modifications of mitochondrial function [24,25,26], to the best of our knowledge, this is the first study to investigate the transcriptional landscape of the gills in this species.

### 2.2. An Insight into the Highly Specialized Transcriptional Profile of the Gills

Due to their crucial role in gas exchange, homeostasis, osmoregulation, and ammonia excretion, the gills represent a tissue of the utmost importance in teleosts and one of the largest in terms of surface exposed to the external environment [32]. Gills are deeply vascularized with blood vessels, which redistribute oxygen and nutrients to the other body districts. Moreover, gills play a crucial function in the fish immune system, as they are in direct contact with the water column and, therefore, with potentially pathogenic microbes present in the marine environment. The immune function of gills is mainly performed by the associated lymphoid tissue (GIALT), which is rich in B cells, T cells, and macrophages/granulocytes; these components provide a significant contribution in terms of genes involved in innate and adaptive immunity in the gills transcriptome [33].

Consistently with these expectations, *C. hamatus* gills displayed a highly distinctive gene expression profile compared with the other four tissues previously analyzed by Song and colleagues (i.e., hearth, skeletal muscle, liver, and head kidney) [26]. A total of 695 transcripts displayed a marked tissue-specificity (Figure 2) and were enriched in functional annotations in line with previously reported physiological functions and morphological features of teleost gills. These included, among others, structural components of gills fibrils (e.g., cytoskeletal keratins and intermediate filaments), components of the gill epithelial barrier (e.g., tight junctions, desmosomes, claudins), and several transcripts involved in mucosal immunity, as reported and discussed in detail in Appendix A.

The results obtained were validated on a larger set of biological samples (i.e., four additional adult individuals) by quantitative real-time PCR (qRT-PCR). Although the results of this analysis need to be considered with caution due to the different year of sampling compared with the individual subjected for RNA-sequencing, they confirmed the strong tissue-specificity of six selected target genes, which displayed high expression values in all biological replicates, and revealed a limited transcription in the other tissues (Appendix A). By exploiting the availability of RNA-seq data from 19 different adult tissues of the red-blooded species *Trematomus bernacchii*, we also verified that the gill-specific genes retained the same pattern of expression in this other notothenioid species (Appendix A).

### 2.3. Evidence of Cold Adaptation in the Transcriptomes of Cryonotothenioidea Gills

Overall, the comparison between the gills gene expression profiles of the six Cryonotothenioidea and the other eight Eupercaria species selected in this study (Appendix A) resulted in a clear-cut separation between these two fish groups (Appendix A).

We applied stringent significance thresholds in the statistical analyses, both in terms of FDR-corrected *p*-value (i.e., <0.05) and in terms of consistency across species (i.e., CS ≥ 36, see Section 3.3), to pinpoint 120 orthologous genes displaying a consistent trend of up-regulation in species adapted to life at sub-zero temperatures. Such differentially expressed genes (DEGs) displayed similar expression profiles in all red-blooded Cryonotothenioidea (*P. borchgrevinki*, *D. mawsoni*, *P. charcoti*, and *Trematomus* spp.). Quite surprisingly, this gene expression signature was also shared by the white-blooded icefish *C. hamatus*, despite the peculiar morpho-physiological features of this species (Figure 3). On the other hand, the gene expression profiles of the 120 candidate orthologs in the sub-Antarctic notothenioid species *Eleginops maclovinus* closely matched those of non-Antarctic species, further reinforcing the likely relatedness of this molecular signature with cold adaptation in Cryonotothenioidea and ruling out the presence of a phylogenetic signature.

In detail, 9 DEGs displayed the maximal CS (i.e., 48) and 43 DEGs have a CS > 40. In some cases, the putative cold adaptation-related transcripts were characterized by extremely significant fold change values, being expressed at several dozen times higher levels in Antarctic species, thereby emerging as the most interesting candidates for in-depth studies. The complete list of DEGs is reported in Appendix A, and the top 20 most significant cold adaptation-related candidate genes in Cryonotothenioidea are reported in Table 1.

### 2.4. Comparative Transcriptome Analysis Reveals Signatures of Cobalamin Deficiency in Cryonotothenioidea

Unexpectedly, the most highly expressed gene in *C. hamatus* gills (nearly 12,000 TPM) encoded a transcobalamin-like protein (TCNL) (Figure 2), implying that more than 1% of the total transcriptional effort of this tissue is employed into the synthesis of an mRNA encoding a protein putatively involved in cobalamin (vitamin B12) binding. The high expression of *TCNL* was shared with other cryonotothenioid species, as this ortholog was the one achieving both the highest CS (48) and the most significant *q*-value (1.81 × 10^−21^) in the comparison of expression profiles with fish species living in temperate environments (Table 1, Figure 4B).

While three different types of cobalamin-binding proteins are present in mammals, i.e., transcobalamin-1 (TCN1), transcobalamin-2 (TCN2), and the gastric intrinsic factor (GIF), previous phylogenetic studies have shown that a single *bona fide* transcobalamin gene is present in teleosts [34,35]. The *C. hamatus* TCNL protein dramatically differs from the canonical teleost transcobalamins, as it lacks the N-terminal α(6)-α(6) barrel domain [36] and only retains the C-terminal α domain, crucial for cobalamin binding-specificity [37]. Considering that *TCNL* orthologs are present in all Eupercaria, we argue these peculiar sequences might have escaped previous investigations as genuine transcobalamins [34,35]. Details about gene/protein architecture and phylogenetic relationship between TCNL and canonical TCNs are provided in Appendix A. The preservation of several conserved residues involved in cobalamin binding in the α domain (Figure 4A) and the presence of a signal peptide suggest that TCNL is released in the bloodstream by endothelial cells present in the gill tissue, possibly functioning as a cobalamin-binding protein.

In mammals, TCN1, GIF, and TCN2 are specifically produced by the salivary glands (TCN1), the parietal cells of the stomach (GIF), and the terminal ileum enterocytes (TCN2), respectively [38]. TCN1 strongly binds ingested B12 vitamin in the oral cavity, protecting this essential vitamin from degradation in the acidic stomach environment and allowing its release in the duodenum. Here, free vitamin B12 is complexed with TCN2, which enables its absorption in the ileum [39]. This complex is quickly degraded upon absorption, allowing the binding between cobalamin and TCN2 for transportation in the portal circulation to the liver. While no expression data are presently available for the icefish digestive tract, we can confirm that the *TCNL* gene lacks significant expression in tissues other than gills in *T. bernacchii* (Appendix A, panel C). To the best of our knowledge, *TCNs* have never been considered as relevant genes in the context of cold adaptation and the gill-specificity of *TCNL* is somewhat puzzling, considering that *TCNs* usually cover their primary physiological function in the digestive tract.

A fascinating hypothesis, which might explain the requirement for an increased cobalamin uptake by an unconventional tissue, is connected with the widespread cestode infections suffered by different Cryonotothenioidea species [40,41,42,43]. Previous histopathological analyses have indeed revealed a remarkably high load of nematode and cestode parasites in *C. hamatus*, up to a few thousand helminths per specimen [44,45]. In detail, it has been demonstrated that this species is the definitive host for 10 different helminth taxa, as well as the intermediate host for another six, with the most severe infections associated with the nematode *Contracaecum osculatum* and, in particular, with tetraphyllidean and diphyllobothridean cestodes [46]. Diphyllobothridean parasites and other tapeworms are not able to synthesize cobalamin de novo [47] and entirely rely on the uptake of dietary cobalamin from their host [48], to the point that their infections in human are typically associated with vitamin B12 deficiency-related pathologies, including megaloblastic anemia [49,50,51]. It is also noteworthy that diphyllobothriasis has profound and deleterious hematological effects in red-blooded fishes, including a decrease in erythrocyte count, due to the fundamental role of folate and vitamin B12 as erythropoiesis cofactors, and a diminished hemoglobin content [52]. However, pathological effects are less harmful in Cryonotothenioidea, due to the increased solubility of gases in fluids and the improved function of their cardio-circulatory system.

Cobalamin is a crucial factor for erythropoiesis and several other key cellular processes, most notably purine and amino acid biosynthesis. Therefore, although Antarctic fish might tolerate hematological alterations, the adoption of appropriate countermeasures would be necessary to maintain the functionality of these housekeeping processes. The hyper-production of TCNL might be part of this molecular rescue system, although the biological significance of this protein in the gills is presently unknown.

Additional transcriptomic evidence supports a condition of cobalamin deficiency in Antarctic fishes, pointing in particular to the presence of hyperhomocysteinemia, a pathological condition usually caused by cobalamin and folate deficiency [53]. In human, this is considered a serious risk factor for the development of cardiovascular diseases due to the blood accumulation of Hcy-thiolactone, a homocysteine metabolite able to impair protein functionality by creating isopeptidic bonds with Lys residues [54]. This dangerous compound is detoxified by the bleomycin hydrolase enzyme (BLMH) [55], whose expression was markedly up-regulated in Antarctic species (*q*-value = 2.18 × 10^−7^, CS = 48; Table 1, Figure 4C).

Another line of evidence pointing in this direction derived from the high expression of methenyltetrahydrofolate synthetase (*MTHFSD*), exceeding by a dozen times the expression of other Eupercarian species (*q*-value = 1.07 × 10^−15^, CS = 48; Table 1, Figure 4D). MTHFSD plays an essential role in the folate catabolism, by remobilizing 5-formylTHF, the only stable form of THF used for storage, into the active folate pool [56]. The constitutive activation of this enzyme might be explained by the continuous demand for active folate to support basic folate-dependent biosynthetic processes. On the other hand, methenyltetrahydrofolate reductase (*MTHFR*) and cystathionine β-synthase (*CBS*), two key enzymes in Hcy detoxification [54], did not show significant trends of up-regulation in Antarctic species.

As a signature of purine biosynthesis impairment, also consistent with cobalamin deficiency, guanosine monophosphate reductase (*GMPR*) was significantly over-expressed in all Cryonotothenioidea (*q*-value = 5.76 × 10^−19^, CS = 48; Table 1, Figure 4E). This enzyme, by converting guanosine monophosphate (GMP) to inosine monophosphate (IMP), has a rescue function, as it allows the re-utilization of free intracellular purine nucleosides.

Overall, the increased expression levels of *TCNL*, *MTHFSD*, *BLMH*, and *GMPR* in the gills of Cryonotothenioidea might represent compensatory mechanisms to counter-balance the heavy loads of helminthic parasites found in the gastrointestinal tract, which on the other hand, might have provided a selective advantage in cold adaptation due to the hematological alterations mentioned above. Under this assumption, the absorption of cobalamin in the gills, or in other tissues, through the interaction between the TCNL-cobalamin complex and a receptor yet to be determined, might mitigate the mitigation of cobalamin deficiency caused by a heavily impaired uptake in other body districts due to widespread dhiplyllobrotriosis. It remains to be investigated whether cobalamin uptake occurs directly from seawater, where this water-soluble molecule is dissolved, or it involves uncharacterized gill-associated cobalamin-producing bacterial symbionts [57].

### 2.5. The High Expression of the Carbonic Anhydrase Genes CA1A and CA4A Explains the High Enzymatic Activity Observed in Antarctic Fish Gills

Carbonic anhydrases (*CAs*) are part of a gene family comprising several members, with specialized functions and diverse tissue localization, and just a few of these enzymes are expressed at biologically relevant levels in the gills of Cryonotothenioidea. Bayesian phylogeny (Figure 5A), which updates previous investigations conducted using a limited number of sequences [58], highlighted the conservation of two paralogous gene copies (named *CA1A* and *CA1B*) in all Eupercaria. These belong to a group homologous to human CA1/2/3, cytosolic enzymes mainly expressed in erythrocytes (CA1 and CA2) and skeletal muscle (CA3). Unlike mammals, Eupercaria possess multiple *CA4-like* genes. Interestingly, two out of the three fish *CA4* isoforms (*CA4A* and *CA4B*) display a domain architecture identical to human *CA4*, whereas the third one contains two consecutive CA domains, in a rearrangement unique to Eupercaria. Phylogenetic inference clearly pointed out that these twin domains are the result of a recent duplication event (Figure 5A). Eupercaria also possess a single *CA5* gene, encoding a mitochondrial CA with essential roles in gluconeogenesis and ureagenesis, as well as a single *CA6-like* gene, which in human is expressed in the salivary gland. The latter encodes a protein containing a C-terminal pentraxin domain (Figure 5A), which appears to be a lineage-specific acquisition. Although transcripts homologous to the other human CA isoforms not mentioned above were seldom detected in the assembled transcriptomes of some species taken into account, they did not reach biologically significant expression levels (TPM < 1) in the gills and were therefore excluded from an in-depth analysis.

Overall, two of the aforementioned CA genes (i.e., *CA1A* and *CA4A*) displayed marked gill-specific expression in *C. hamatus*, reaching extremely high expression levels (~4000 and ~9000 TPM, respectively, see Figure 2). This observation was confirmed by qRT-PCR and in silico analyses in *C. hamatus* (Appendix A) and *T. bernacchii* (Appendix A), respectively. Although *CA4C* also displayed gill-specificity, it was expressed at values lower than 5% compared to the other two isoforms. *CA4B* and the mitochondrial isoform *CA5* were expressed at moderate levels in all tissues (Figure 5B). *CA1B*, expressed at moderate levels in gills, was also found in head kidney, and *CA6* was found in heath and skeletal muscle.

*CA1A* (CS = 46, *q*-value = 2.45 × 10^−5^) was included among the most significant orthologs involved in cold-adaptation, due to its high expression in Cryonotothenioidea compared with other Eupercaria (Table 1). Although *CA4A* and *CA4B* were not included in the DEG shortlist due to their relatively low CS (33 and 32, respectively), they also displayed a general tendency of higher expression in the gills of Antarctic species, as evidenced by the visual representation of the cumulative expression of all CA variants (Figure 5C). Overall, CAs contributed to 4.68 ± 2.01% of the global gill transcriptional effort in Cryonototheniodea, compared to 1.25 ± 0.87% in other Eupercaria. This is in line with the previously reported higher CA enzymatic activity in the gills of *C. hamatus* and *T. bernacchii* compared with non-Antarctic species [59,60]. Altogether, these data point out that the high CA enzymatic activity associated with the gills of Antarctic fish is likely linked with the increased expression level of three CA genes shared by all Eupercaria (i.e., the membrane-bound isoforms *CA4A* and *CA4C*, and the cytosolic isoform *CA1A*), and not merely ascribable to an improved catalytic activity of these enzymes at low temperatures [59,60].

The reasons underpinning the intense production of CA in this tissue in the polar environment may be intertwined with the alterations of the gas transport system documented in Cryonotothenioidea, and in icefishes in particular. It is now well-recognized that CAs play a crucial role in fish gills, where they catalyze the hydration of CO_2_ to HCO^3−^ and H^+^, affecting the concentration gradient for gas exchange. Moreover, this reaction is at the basis of blood acid-base regulation [61] and therefore, in species with a reduced blood oxygen-carrying capacity, CAs are likely to cover an essential function in the restoration of the blood acid/base homeostasis following swimming exercise [10]. The relevant gill activity of CAs in Antarctic notothenioids might explain the observation that they show a remarkable respiratory increase instead of the expected metabolic acidosis following strenuous activity [62].

In light of these observations, the type and subcellular localization of CA involved in efficient CO_2_ conversion in the gills is of particular interest. CA4 isotypes are particularly relevant in the context of the gill tissue, since mammalian CA4 is typically expressed at high levels in lung epithelium, where its protein product is anchored to the luminal side of the pulmonary membrane. Other authors have previously described the expression of a membrane-bound CA isoform, accessible to the blood plasma, in the gills of the icefish *Champsocephalus gunnari*, hypothesizing its participation in CO_2_ excretion [63]. However, the partial sequence reported by Harter and colleagues belongs to the CA4B group, most likely covering just a minor role in the global CA activity in the gills of Cryonotothenioidea (Figure 4C). Moreover, a previous study has demonstrated that, in the white-blooded notothenioid *Chaenocephalus aceratus*, a membrane-bound CA4-like protein only accounted for less than 3% of the total CA activity observed in the gills, whereas the remaining 97% was ascribable to a cytosolic isoform [64]. In light of this information, the most likely candidate isoform for the extraordinary CA enzymatic activity in the gills of Antarctic fishes would be CA1A.

It is also worth mentioning that, despite the high levels of expression of CAs in the gills of both white- and red-blooded Cryonotothenioidea, no CA activity is detectable in *C. hamatus* blood [59]. This finds an explanation in the lack of functional red blood cells, which in vertebrate contain an appropriate amount of CA1, as well as in the presence of an endogenous plasma CA inhibitor in the blood of Channichthyidae [59]. We hypothesize that CA1B, only expressed at low levels in *C. hamatus* gills and head kidney (Figure 5B), could represent the cytosolic CA isoform responsible of the CA activity of teleost red blood cells. Altogether, these observations support the development of experiments dedicated at an improved study of CA function in Antarctic fish, as these enzymes emerge as primary players in the physiological adaptations to cold.

### 2.6. Constitutive Expression of Chaperones, Co-Chaperones, and the Proteasome Machinery

Freezing temperatures do not just present a significant challenge for metabolic adaptations, but also for fundamental cellular processes such as protein synthesis and folding. Previous studies have evidenced that Antarctic fish species have developed specific strategies to assist protein folding at low temperatures, such as an increased affinity of the cytosolic chaperone CCP to its client proteins [65] and the shift from inducible to constitutive expression of HSPs [27,66,67]. While the high expression of molecular chaperones has been previously evidenced in *Dissostichus mawsoni* [68], here we provide additional insights concerning the enhanced expression of several different molecular chaperones, as well as of components of the proteasome machinery for the degradation of unfolded proteins.

Namely, *FKBP3*, a peptidyl-prolyl cis-trans isomerase which catalyzes the formation of disulfide bonds during oxidative folding [69], was up-regulated in all Cryonotothenioidea compared to non-Antarctic fishes (*q*-value = 3.5 × 10^−4^, CS = 41 Figure 6B). Similarly, the peptidylprolyl isomerase D (*PPID*), an *HSP90* co-chaperone which accelerates protein folding, and *PARP16*, a key regulator of unfolded protein response in the endoplasmic reticulum, were both upregulated (*q*-value = 1.08 × 10^−7^/CS = 42 and *q*-value = 0.028/CS = 44, respectively; Figure 6B). Although with relevant fluctuations across species, also the warm inducible protein 65 (*WAP65*) was significantly overexpressed in Cryonotothenioidea (*q*-value = 2.75 × 10^−5^, CS = 46) [70,71].

In agreement with the role of proline as a chemical thermoprotectant [72], two enzymes involved in the biosynthesis of this amino acid were significantly up-regulated, namely the pyrroline-5-carboxylate reductase *PYCR3* (*q*-value = 1.23 × 10^−10^, CS = 40) and the ornithine aminotransferase *OAT* (*q*-value = 6.09 × 10^−6^, CS = 40) (Figure 6B).

Protein glycosylation is another molecular mechanism used to assist protein folding and improving their stability, by altering folding kinetics [73,74]. Interestingly, we can report that some enzymes involved in this process were up-regulated in Antarctic notothenioids, such as *OSTC*, a subunit of the oligosaccharyltransferase complex involved in the N-glycosylation of nascent polypeptidic chains (*q*-value = 1.36 × 10^−8^, CS = 48, Figure 6B).

While the high constitutive expression of chaperones had already been reported in Antarctic fish [27], we can report here for the first time that a number of genes encoding structural components of the proteasome follow a similar trend of expression, possibly as a result of the low rate of successfully folded newly synthesized proteins. Indeed, previous reports showed that the rate of functional newly synthesized proteins is only in the range of 15–20% in polar marine ectotherms. In contrast, both the levels of protein ubiquitination and RNA to protein ratios are much higher than those observed in species living in temperate environments [75]. In detail, our data reveal a significant over-expression of several proteasome subunits, in particular those pertaining to the regulatory particle 19S (Figure 6A).

Altogether, this suggests that, in spite of the significant enhancement of unfolded protein response in the ER, a significant fraction of the newly synthesized proteins fail to fold properly, being therefore directed to proteasomal degradation. This conclusion is supported by the recent report of strong proteasome activity in *T. bernacchii*, which has been correlated with a high rate of protein degradation [76].

The numerous signatures of alteration of nascent polypeptide glycosylation, unfolded protein response, and proteasomal activity shared by the four Antarctic species point out that these biological pathways are commonly affected in all Cryonotothenioidea, irrespective of species-specific morpho-physiological adaptations, confirming the similar energetic cost of protein synthesis reported for red- and white-blooded nototheniods [77].

### 2.7. Other Remarks and Conclusions

The genes described in detail in the previous sections only represent a fraction of those that met the criteria we arbitrarily set (Appendix A), while many others either have an unknown function or show no apparent connection to cold adaptation. We also need to stress out that many genes showing significant *q*-values, but moderate consistency score may be involved into cold tolerance as well, although not specifically in the gill tissue, or may be otherwise subject to highly significant inter-specific variations of expression levels. Undoubtedly, several such cases might require additional targeted investigations in the future. One of the most evident, but still unexplained, biological processes whose upregulation in Cryonotothenioidea gills is supported by the identification of multiple genes is the regulation of vesicular trafficking, clathrin-mediated endocytosis in particular. In these respects, *TWF1* (*q*-value = 2.01 × 10^−9^, CS = 38), *TMED10* (*q*-value = 1.73 × 10^−5^, CS = 43), *VPS53* (*q*-value = 1.54 × 10^−8^, CS = 37), *EHD3* (*q*-value = 1.38 × 10^−18^, CS = 43), *FCHO2* (*q*-value = 7.05 × 10^−13^, CS = 41), and *RHOV* (*q*-value = 1.43 × 10^−13^, CS = 47) are all relevant examples. Moreover, several features unique to *C. hamatus* (and, possibily, to other icefishes), but not shared with red-blooded notothenioids may have been missed by our comparative approach. Among these, an observation of high interest from an immunological point of view is represented by the complete lack of expression of the immunoglobulin Tau heavy chain gene, which has a role of primary importance in fish mucosal immunity and is generally highly expressed in this tissue [78]. This would reinforce preliminary observations that point towards a possible gene loss event in Channichthyidae, that would represent just one of the several peculiar features in the evolution of the immunoglobulin heavy chain genes of Antarctic fishes [79].

This work provides the first large scale comparative transcriptomics analysis carried out in the gills of Cryonototheniodea, with the aim to identify genes significantly up-regulated in Antarctic fish species compared to temperate Eupercaria. Among the 120 selected orthologous genes, we can report several previously unreported molecular players, which might play an essential role in the adaptation to the freezing Antarctica environment. Surprisingly, such alterations were not limited to Channichthyidae, but they were shared by all the analyzed cryonotothenoids, regardless of the presence or absence of functional erythrocytes.

These data revealed an unexpected alteration of biological processes linked to cobalamin metabolism, evidencing molecular signatures of hyperhomocysteinemia. These observations open several interesting questions concerning the possible connection between this pathological condition and the heavy loads of parasites observed in cryonotothenioids, within the evolutionary frame that led to the development of a peculiar cardio-circulatory system in Antarctic teleosts, characterized by the lack of hemoglobin in Channichthyidae.

## 3. Materials and Methods

### 3.1. Chionodraco Hamatus Transcriptome de Novo Assembly and Annotation

An adult *C. hamatus* specimen of unspecified sex, weighting about 150 g, was collected with the aid of a small demersal gillnet, as established by the Commission for the Conservation of Antarctic Marine Living Resources (CCAMLR), which was left in the water for four hours, i.e., a timing which was selected to maximize the chances of catching live specimens, minimizing the impact of sampling on non-target marine fauna. The catch was made close to the Mario Zucchelli base (Ross Sea 74°41′42″ S, 164°07′23″ E) during the Antarctic summer (January–February) of 2009, within the frame of the Italian National Program of Research in Antarctica (PNRA). Upon collection, the fish was immediately transported to the base in an aerated bin and placed in a tank with running natural seawater, kept at the same temperature recorded in natural environment (i.e., −2 °C). After one week of acclimatization, which allowed the fish to recover from the stress linked with the capture, the specimen was lethally anesthetized with 1 mg × mL^−1^ tricaine methanesulfonate (Sigma, Saint Louis, MO, USA) and the gills were immediately dissected and placed in RNAlater solution (Thermo Fisher Scientific, Waltham, MA, USA). RNA extraction was performed with TriPure (Roche, Basel, Switzerland) following the manufacturer’s instructions. The quality of the extracted RNA was evaluated with a Bioanalyzer 2100 instrument (Agilent Technologies, Santa Clara, CA, USA) to ensure that the RNA integrity number was higher than 9 and that no DNA, protein, and phenol contamination was present, as assessed by 260:230 and 260:280 absorbance rates by spectrophotometric analysis. The extracted RNA was shipped to the Institute of Applied Genomics (Udine, Italy) for the preparation of an Illumina TruSeq library (Illumina, San Diego, CA, USA) for paired-end sequencing on a single lane of an Illumina HiSeq2000 platform, with a 2 × 100 paired-end strategy. The raw reads, deposited at the NCBI SRA database under the accession ID SRX2181658 (Bioproject: PRJNA343733), were imported in the CLC Genomics Benchwork v.20 (Qiagen, Hilden, Germany) and trimmed to remove residual adapters and low-quality nucleotides (the base caller quality threshold was set at 0.05; resulting reads shorter than 75 nt were discarded). The transcriptome was generated with the de novo assembly tool, with the following settings: *word size* and *bubble size* were automatically estimated, minimum contig length was set to 250 nt, and scaffolding was performed to fully exploit pair-end information.

Sequencing reads were mapped to the reference assembly to obtain raw read counts, allowing the calculation of expression levels. The mapping parameters were set as follows: mismatch cost = 3; deletion cost = 3; insertion cost = 3; length fraction = 0.75; similarity fraction = 0.98. Gene expression values were calculated as transcript per million (TPM). Contigs displaying an expression level lower than 1 TPM were discarded, to enable the removal of contaminants and poorly assembled sequences with low annotation rates [80,81].

The assembly was functionally annotated with the Trinotate pipeline (version 3.2.1, https://trinotate.github.io), which assigned to each contig Pfam conserved protein domains and Gene Ontology (GO) terms. The completeness and fragmentation of the assembled transcriptome was assessed with BUSCO v.3, based on the Actinopterygii-specific set of orthologous sequences odb9 [82].

### 3.2. Assessment and Validation of Tissue-Wide Gene Expression Profile

The available transcriptome data obtained from other adult tissues of the same species were recovered from the NCBI-SRA database. Namely, the following tissues were selected: heart (SRX1542182), skeletal muscle (SRX1542183), liver (SRX1538870), and head kidney (SRX1067737, SRX1067738, SRX1067739) [25,26]. Trimmed sequencing reads were mapped to the reference transcriptome as described in Section 3.1. Log_10_ transformed TPM values were used to build a heat map, based on the hierarchical clustering of a gene subset (those achieving a minimum TPM value of 1000 in at least one tissue). Gill-specific transcripts were identified using arbitrarily set thresholds of expression. In detail, only the contigs falling within the top 10% most highly expressed in the gills and, at the same time, showing fold-change > 10 in all the pairwise comparisons with the four other tissues were selected. The gill-specific contig list was subjected to a hypergeometric test on GO terms and Pfam domains annotations to identify the biological processes (BP), molecular functions (MF), cellular component (CC), and gene families most prominently associated with the function of this tissue.

Considering the origin of sequence data from a single specimen, and the potential existence of inter-individual variation in gene expression related to sex, age, and genetic factors, we confirmed the gill specificity of six target genes, selected due to their high expression in this tissue (Table 1), by qRT-PCR, performed on multiple tissues collected from additional biological replicates. In detail, four *C. hamatus* adult specimens were collected close to the Mario Zucchelli Station in November 2017 with the aid of nets, anesthetized and dissected as explained in Section 3.1, collecting gills, brain, head kidney, skeletal muscle and liver tissues. Upon RNA extraction, cDNAs were prepared with a qScript™ Flex cDNA Synthesis Kit (Quanta BioSciences Inc., Gaithersburg, MD, USA) based on the manufacturer’s instructions. 1 μL of 1:20 diluted cDNA was added to the 15 μL PCR reaction mix, containing 7.5 μL SsoAdvanced SYBR Green Supermix (Bio-Rad, Hercules, CA, USA), 0.2 μL of each of the 10 μM primers and water. The reaction was carried out on a C1000-CFX96 platform (Bio-rad, Hercules, CA, USA), with a two-step protocol comprising 40 amplification cycles at 95° (10 s) and 60° (20 s), performed after an initial denaturation step at 95 °C for 2 min. At the end of the PCR reaction, a melting curve analysis (from 65 to 95 °C) was carried out to assess the specificity of amplification. The levels of expression for the target genes, normalized on two housekeeping genes, were calculated based on the Ct method, taking into account the standard deviation provided by three technical replicates. The forward and reverse primers for the six target genes and the two housekeeping genes, namely the elongation factor 1 alpha (*EF1A*) and the 40S ribosomal protein S7 (*RPS7*) (Appendix A), were designed with Primer3Plus (https://primer3plus.com/), aiming at an average amplicon size of 120 nt. The two housekeeping genes were selected based on: (i) stability of expression in *C. hamatus*, evaluated by a low standard deviation of TPM values in RNA-seq data (*RPS7*), or (ii) by previous validation as suitable targets for RT-PCR in other teleost species (*EF1A*) [83,84].

Tissue specificity of expression of specific gene targets was further validated in silico in a cross-species comparison, by analyzing the available RNA-seq datasets from an unpublished study carried out in *T. bernacchii* (Bioproject accession: PRJNA471228). Trimmed reads were mapped to the reference transcriptome as described in Section 3.1 to calculate TPM gene expression values in 19 different adult tissues.

### 3.3. Comparative Gene Expression Analysis

Selected transcriptome datasets obtained from gills of 13 species pertaining to the taxon Eupercaria (Appendix A) were downloaded from the NCBI-SRA database. For each species, raw reads were processed, trimmed, and de novo assembled as described in Section 3.1.

Five out of the species considered, i.e., *T. bernacchii*, *T. newnesi*, *P. borchgrevinki*, *P. charcoti*, and *D. mawsoni*, belong to the Cryonotothenioidea taxon and are adapted to the Antarctic environment, even though none of them belongs to the Channichthyidae family. The other eight species were heterogeneous in terms of range of distribution, salinity, and temperature preference and included the non-Antarctic notothenioid *E. maclovinus*.

The inter-species comparison of gene expression profiles is a relevant challenge in comparative genomics. These analyses are indeed hampered by inherent difficulties linked to different quality of genome/transcriptome assembly and annotation, the activity of transposable elements, the extent of lineage-specific gene gains, losses, and duplications which overall complicate orthology inference and may significantly unbalance the calculation of gene expression levels in different taxa [85]. This issue can be only tackled with heuristic approaches, such as using a gene subset shared by all the taxa taken into account for the calculation of gene expression levels. Although this strategy is inevitably limited to the analysis of single-copy orthologous genes, the obtained expression levels can be effectively compared across species [80,86].

In the present study, we applied a similar approach to evaluate whether the gills of Antarctic fish species display any significant molecular signatures of adaptation to freezing temperatures compared to non-Antarctic Eupercaria. We used a set of single-copy orthologous genes conserved across all Actinopterygii, extracting from each of the target transcriptomes the 4584 actinopterygiian BUSCOs included odb9, to which we manually added 15 additional orthologs with particular relevance in the context of cold adaptation. This set of orthologous genes was used for the calculation of gene expression levels through the mapping of the original trimmed reads using Kallisto v.0.43.0, with 100 bootstrap replicates [87]. Whenever multiple RNA-seq libraries were available for a given species, gene expression data were averaged across samples. Whenever “missing orthologs” were found, this was considered evidence of lack of expression, and therefore the given gene was assigned a TPM value = 0. The gene expression matrix obtained was subjected to a statistical analysis of differential gene expression, to identify differentially expressed genes (DEGs) with expression values significantly higher in Cryonotothenioidea species compared to other Eupercaria. This analysis was carried out with the Sleuth platform (https://github.com/pachterlab/sleuth, accessed on July 2016). The criteria used for the detection of differentially expressed genes (up-regulated in Cryonotothenioidea) was based on a *q*-value < 0.05 in the comparison between the two group species. In addition, to take into account the presence of outlier species, as well as of possible errors in the calculation of gene expression levels due to misassembly and undetected chimeric contigs in some species, we assigned to each orthologous group a consistency score (CS, ranging from 0 to 48). CS was calculated based on the number of pairwise comparisons between each Antarctic and non-Antarctic Eupercaria species where a fold-change >1.5 was attained. The final set of cold adaptation-related candidate genes was selected based on a consistency score ≥36 (i.e., displaying a significantly higher expression value in ≥75% pairwise comparisons) and a *q*-value < 0.05.

### 3.4. Phylogenetic Analyses

The amino acid sequences of carbonic anhydrases from the 14 Eupercaria species were predicted from the corresponding assembled sequences. Based on the outcome of comparative gene expression analysis, only sequences expressed at significant levels in the gills of all Cryonotothenioidea (TPM > 10) were considered. For classification purpose, the sequences of human CAs were also included. The sequence of the human carbonic anhydrase related protein (CARP) and teleosts orthologous sequences were used to provide an outgroup for tree rooting.

The portion of the sequence corresponding to the CA domain, detected based on the coordinates of Pfam domain, was extracted from each sequence. In the case of teleost proteins displaying two CA domains, the sequence of both domains was extracted. Sequences derived from highly fragmented contigs (whose length was lower than 75% of the expected domain length) were discarded. The resulting sequences of the CA domains were aligned with MUSCLE [88]. The multiple sequence alignment was then subjected to a ModelTest-NG [89] to detect the WAG+G+I+F model as the best-fitting model of molecular evolution for the CA sequence set. MrBayes v.3.2.7a [90] was used to run a Bayesian phylogenetic analysis, implementing the aforementioned model until the reaching of standard deviation of split frequencies <0.05 (i.e., 500,000 generations). Run convergence was assessed by the reaching of ESS values >200 for all estimated parameters, evaluated with a Tracer v.1.7.1 analysis. A similar approach was also used with TCN and TCNL proteins, as detailed in Appendix A.

## Figures and Tables

**Figure 1 ijms-22-01812-f001:**
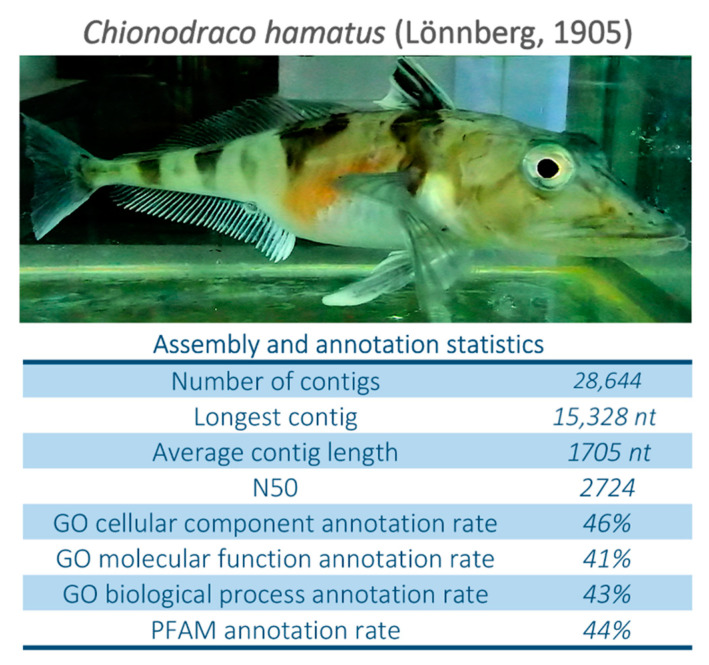
External morphology of an adult *Chionodraco hamatus* specimen, with the assembly and annotation statistics obtained for gill transcriptome.

**Figure 2 ijms-22-01812-f002:**
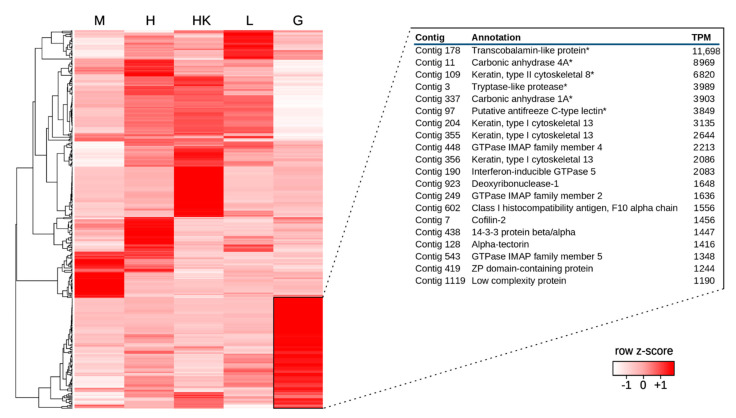
Hierarchical clustering of a set of representative genes (achieving a TPM value > 1000 in at least one tissue), based on the gene expression levels, i.e., Log10(TPM+1), of skeletal muscle (M), hearth (H), head kidney (HK), liver (L), and gills (G). Genes were clustered with an average linkage method, based on Pearson distances. Colors represent the Z-scores, calculated for each row based on the maximum and minimum gene expression level observed. Gills-specific genes are boxed, and the top 20 most highly expressed gill-specific protein-coding transcripts, defined by a fold change value > 10 in all the pairwise comparisons with the other available tissues, are shown in a table. TPM = Transcripts Per Million. * target genes for validation by qRT-PCR, see Appendix A.

**Figure 3 ijms-22-01812-f003:**
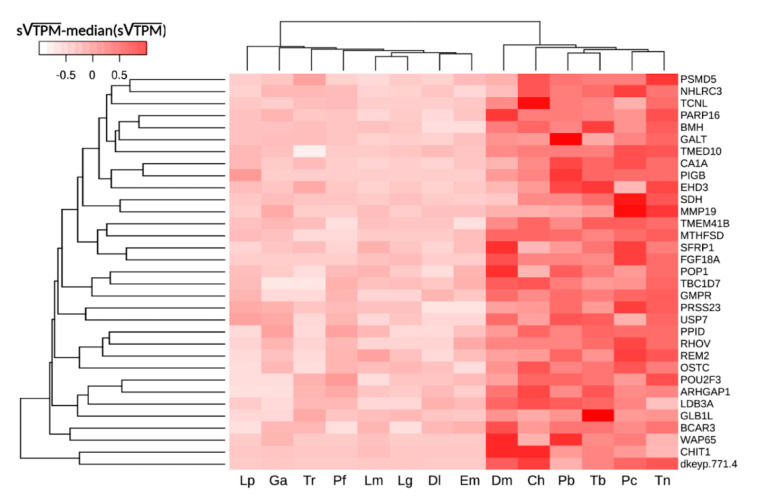
Hierarchical clustering of the six Cryonotothotenioidea and eight additional Eupercaria species considered in this study, based on gills gene expression profiles. An average linkage clustering method was used, based on Euclidean distance, which considered the expression levels of the 33 orthologous genes displaying the highest consistency score of differential expression between the two groups (≥42). For graphical representation, each gene’s square root-transformed TPM values were first standardized on the maximal level observed across species, and subsequently subtracted with the median standardized square root-transformed TPM value of all species. Ch: *Chionodraco hamatus*; Dl: *Dicentrarchus labrax*; Dm: *Dissostichus mawsoni*; Em: *Eleginops maclovinus*; Ga: *Gasterosteus aculeatus*; Lg, *Ljutanus guttatus*; Lm: *Lateolabrax maculatus*; Lp: *Larimichthys polyactis*; Pb: *Pagothenia borchgrevinki*; Pc: *Parachaenichthys charcoti*; Pf: *Perca fluviatilis*; Tb: *Trematomus bernacchii*; Tn: *Trematomus newnesi*; Tr: *Takifugu rubripes*.

**Figure 4 ijms-22-01812-f004:**
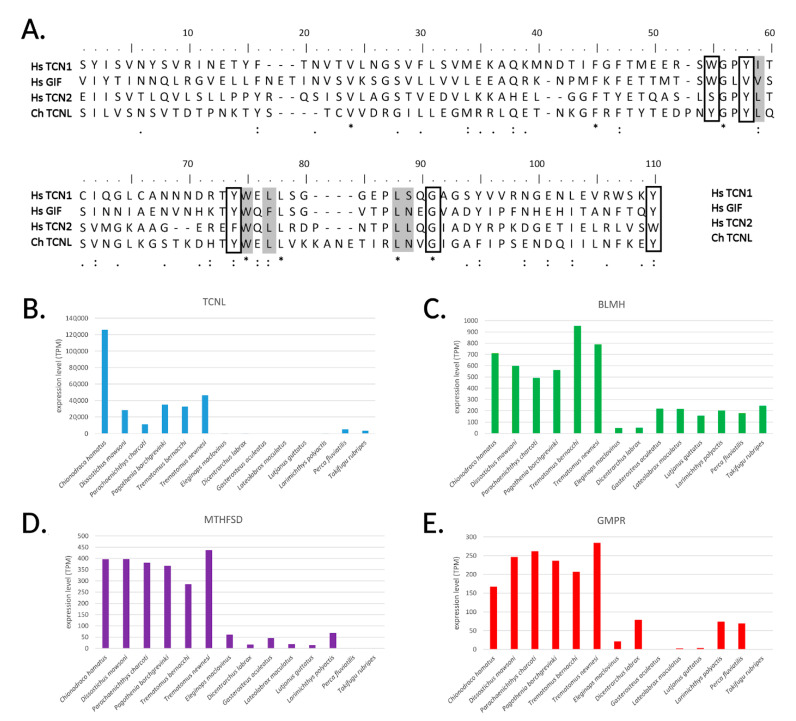
Panel (**A**): Multiple sequence alignment of C-terminal α domain from the *C: hamatus* transcobalamin-like protein (TCNL) and the human transcobalamin-1, -2 and gastric intrinsic factor. Residues involved in hydrophobic interactions with cobalamin are shown in a box; residues involved in forming hydrogen bonds with cobalamin are shaded (from Wuerges et al., 2007). Panels (**B**–**E**): comparative expression levels of transcobalamin-like (*TCNL*), bleomycin hydrolase (*BLMH*), methenyltetrahydrofolate synthetase (*MTHFSD*), and guanosine monophosphate reductase (*GMPR*) orthologs in Cryonotothenioidea and non-Antarctic Eupercaria.

**Figure 5 ijms-22-01812-f005:**
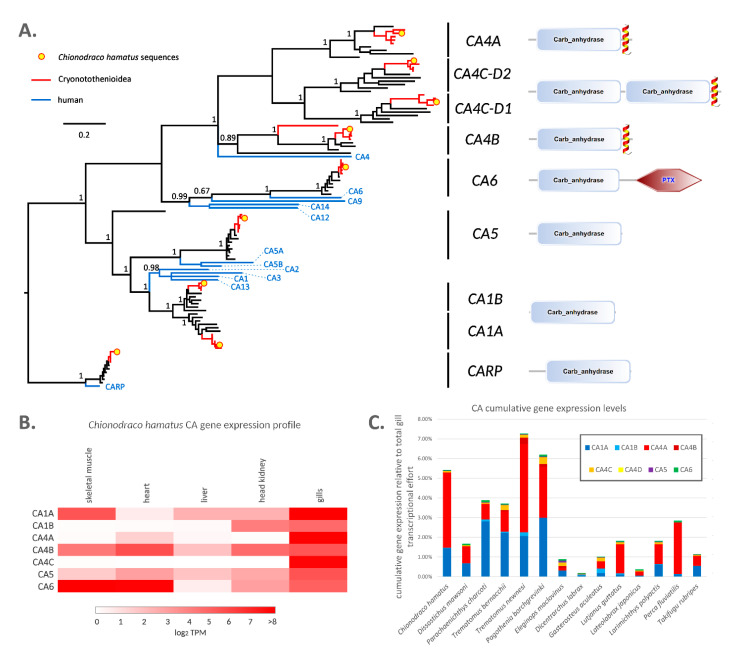
Panel (**A**): Bayesian phylogeny of carbonic anhydrases from Eupercaria, based on a WAG+G+I+F model of molecular evolution and on the multiple sequence alignment of the CA domain (the two domains of CA4C proteins were both included). For methodological details, see the Materials and Methods section. The carbonic anhydrase-related proteins (CARP) clade was used as an outgroup to root the tree. Posterior probability support values for the primary nodes are shown. Nodes supported by poor posterior probability values (<0.5) were collapsed. Sequences from Cryonotothenioidea are indicated with red branches, and those from *C. hamatus* are marked with a yellow circle. Sequences from *Homo sapiens*, included to allow orthology assessment, are indicated with blue branches. The domain architecture of each CA group is schematically displayed; transmembrane domains are indicated by a helix. PTX: pentraxin. Panel (**B**): heat map summarizing gene expression levels of carbonic anhydrases in different tissues of *C. hamatus*. Gene expression levels are shown as log_2_TPM. Panel (**C**): cumulative gene expression levels of all carbonic anhydrase genes in the 14 Eupercaria species considered in the present study. Gene expression levels are shown as the relative contribution to the total gene expression of the gill tissue, calculated on the BUSCO orthologous gene set.

**Figure 6 ijms-22-01812-f006:**
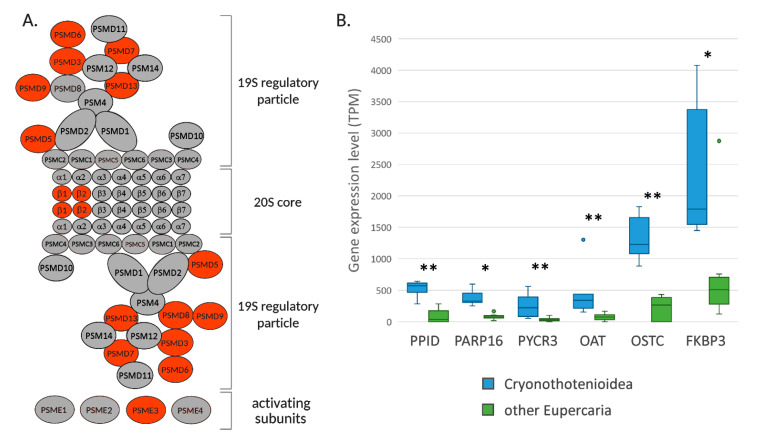
Panel (**A**): schematic representation of proteasome subunits, indicating those showing increased expression levels in Cryonotothenioidea in red. Panel (**B**): comparative expression levels of peptidylprolyl isomerase D (PPID), poly(ADP-ribose) polymerase family, member 16 (PARP16), pyrroline-5-carboxylate reductase (PYCR3), ornithine aminotransferase (OAT), oligosaccharyltransferase complex non-catalytic subunit (OSTC), and FK506 binding protein 3 (FKBP3) and orthologs in Cryonotothenioidea and non-Antarctic Eupercaria. * = *q*-value < 0.05; ** = *q*-value < 1 × 10^−5^.

**Table 1 ijms-22-01812-t001:** Top 20 most significantly up-regulated orthologous genes in Cryonototheniodea, compared with non-Antarctic Eupercaria. CS: consistency score; qval: FDR-corrected *p*-value; FC: fold change.

Gene	Description	CS	Qval	FC	Putative Function
*TCNL*	transcobalamin-like protein	48	1.81 × 10^−21^	31.20	cobalamin carrier protein
*dkeyp-77h1.4*	si:dkeyp-77h1.4	48	1.02 × 10^−20^	494.00	unknown
*GMPR*	guanosine monophosphate reductase	48	5.76 × 10^−19^	9.96	purine biosynthesis
*MTHFSD*	methenyltetrahydrofolate synthetase domain containing	48	1.07 × 10^−15^	11.34	tetrahydrofolate metabolism regulator
*CHIT1*	chitotriosidase-I	48	3.45 × 10^−14^	107.97	defense against nematodes
*FGF18A*	fibroblast growth factor 18a	48	1.85 × 10^−10^	46.61	cell proliferation regulator
*OSTC*	oligosaccharyltransferase complex subunit	48	1.36 × 10^−8^	5.38	N-glycosylation of nascent polypeptides
*BLMH*	bleomycin hydrolase	48	2.18 × 10^−7^	4.52	homocysteine thiolactone detoxyfication
*TMEM41B*	transmembrane protein 41B	48	1.46 × 10^−5^	6.03	autophagosome formation
*RHOV*	ras homolog family member V	47	1.43 × 10^−13^	7.20	actin cytoskeleton control
*PIGB*	phosphatidylinositol glycan anchor biosynthesis, class B	46	3.95 × 10^−43^	30.20	GPI-anchor biosynthesis.
*CA1A*	carbonic anhydrase 1a	46	2.45 × 10^−5^	8.71	CO_2_ excretion, cellular pH regulation
*WAP65*	Warm-temperature acclimation related 65 kDa protein	46	2.75 × 10^−5^	22.85	stress response gene
*ARHGAP1*	Rho GTPase activating protein 1	45	2.49 × 10^−10^	7.37	GTPase activator
*LDB3*	LIM domain binding 3	44	9.36 × 10^−15^	8.38	scaffolding protein
*SFRP1*	frizzled-related protein	44	1.47 × 10^−8^	9.20	Wnt signaling modulator
*MMP19*	matrix metallopeptidase 19	44	8.42 × 10^−8^	6.05	extracellular matrix remodeling
*NHLRC3*	NHL repeat containing 3	44	5.65 × 10^−7^	3.83	unknown
*GALT*	galactose-1-phosphate uridylyltransferase	44	2.81 × 10^−3^	4.44	galactose catabolism
*PARP16*	poly (ADP-ribose) polymerase family, member 16	44	2.83 × 10^−2^	3.56	unfolded protein response

## Data Availability

The raw reads generated by RNA-seq have been deposited at the NCBI SRA database under the accession ID SRX2181658, and linked with the Bioproject PRJNA343733.

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
