# Peer review of "Cold Adaptation in Antarctic Notothenioids: Comparative Transcriptomics Reveals Novel Insights in the Peculiar Role of Gills and Highlights Signatures of Cobalamin Deficiency"

_ijms, 2021, doi:10.3390/ijms22041812_

Round 1

Reviewer 1 Report

The manuscript about the transcriptomic analysis of gills Antarctic notothenioids shows new information and several genes being up-regulated in Antarctic species. Α detailed analysis of the results has been described, and overall the manuscript is well written. I do have some issues about the methodology and some other minor changes. Please check below

Introduction:

As stated by the authors the taxonomy is still being challenged, but authors did choose to specify that Cryonotothenioidea has 5 families. Please refer to reference 6 were this taxonomy is being explained. 

Line 106: Where is figure 1A? 

Line 110: Please correct out to our

Methodology:

Do authors have the approval of the ethics committee for the use of animals?

Section 3.1 - Only a single adult fish has been used for the transcriptome which is not representative, there might be biological differences (although authors are aware if this issue and highlighted this several times throughout the manuscript and supplemental data). Even so, a conclusion stating that it fully confirmes the sequencing data is too risky, please change accordingly in Supplementary data note 2. 

Collection by nets does collect other animals, and is invasive for the environment, how did authors take care of this? Many details are missing in this section about the transportation of the fish to the base and keeping it in a tank. Was oxygen supplied to the water? was natural water used? How long was the fish kept in the tank before sampling? 

Section 3.2 - 8 years later 4 adult fish were sampled which were used to confirm target genes. In 8 years several differences might have changed the expression, which is also stated in supplemental data. I believe that this information should also be in the manuscript itself as this is crucial information.

Line 577: please correct section 2.1 to 3.1

Results:

Line 115: Where is Figure 1B?

Line 166-167: Five additional adult individuals? In the methodology , line 575, it was stated that four specimens were collected.

Supplementary note 2: Please correct RT-PCT into RT-PCR

Line 313: please correct gastrointestinal

Conclusion:

Line 499: please correct preliminary

Line 500: please correct peculiar

Author Response

Reviewer #1

The manuscript about the transcriptomic analysis of gills Antarctic notothenioids shows new information and several genes being up-regulated in Antarctic species. Α detailed analysis of the results has been described, and overall the manuscript is well written. I do have some issues about the methodology and some other minor changes. Please check below

#Thank you for the positive assessment of our work. We tried to address all the comments made by the reviewer, as detailed below. All changes are marked in the main text with the “track changes” mode.

Introduction:

As stated by the authors the taxonomy is still being challenged, but authors did choose to specify that Cryonotothenioidea has 5 families. Please refer to reference 6 were this taxonomy is being explained

#Thank you for this suggestion. We added reference 6 here and clarified the sentence as follows, to expand the definition and specify that three additional families of non-Antarctic notothenioids exist:

“From an evolutionary and taxonomical perspective, Cryonotothenioidea include five out of the eight families recognized within Notothenioidea according to the World Register of Marine Species, i.e. Harpagiferidae, Artedidraconidae, Bathydraconidae, Channichthyidae and Nototheniidae [6], whereas non-Antarctic notothenioids include Bovichtidae, Eleginopsidae and Pseudaphritidae. However, it is worth noting that recent molecular investigations have challenged the monophyly of some of these clades [16].”

Line 106: Where is figure 1A? 

#Thank you port pointing this out. Figure 1 was originally planned to have a different structure, being divided in multiple panels, and we forgot to remove the reference to panel A here. This has been fixed.

Line 110: Please correct out to our

#Fixed

Methodology:

Do authors have the approval of the ethics committee for the use of animals?

#Yes, please refer to the “Institutional Review Board Statement” section at the end of the text, that the reviewer may have missed due to its placement, which read: “The sample collection and animal research conducted in this study comply with Italy's Ministry of Education, University and Research regulations concerning activities and environmental protection in Antarctica and with the Protocol on Environmental Protection to the 137 Antarctic Treaty, Annex II, Art. 3.”

We also added the following additional details:

All experiments have been performed in accordance with the U.K. Animals (Scientific Procedures) Act, 1986 and associated guidelines, EU Directive 2010/63/EU and Italian DL 2014/26 for animal experiments.

“All the activities on animals performed during the Italian Antarctic Expedition are under the control of a PNRA Ethics Referent, which acts on behalf of the Italian Ministry of Foreign Affairs. In particular, the required data are the following. Project identification code: PNRA16_00099. Name of the ethics committee or institutional review board: Italian Ministry of Foreign A airs. Name of PNRA Ethics Referent: Dr. Carla Ubaldi, ENEA Antarctica, Technical Unit (UTA).”

Section 3.1 - Only a single adult fish has been used for the transcriptome which is not representative, there might be biological differences (although authors are aware if this issue and highlighted this several times throughout the manuscript and supplemental data). Even so, a conclusion stating that it fully confirmes the sequencing data is too risky, please change accordingly in Supplementary data note 2. 

#We accept this criticism. We added the following sentence to Supplementary Data Note 2:

“However, we  cannot exclude that the marked tissue-specificity of some of the genes whose pattern of expression could not be validated with this approach are linked with factors affecting the physiology of the single individual subjected to RNA-sequencing, such as age, sex, feeding status and presence of undocumented pathological conditions.”

Collection by nets does collect other animals, and is invasive for the environment, how did authors take care of this? Many details are missing in this section about the transportation of the fish to the base and keeping it in a tank. Was oxygen supplied to the water? was natural water used? How long was the fish kept in the tank before sampling? 

#For the sampling, a small (1x80 m) demersal gillnet was used as established by the Commission for the Conservation of Antarctic Marine Living Resources (CCAMLR). Furthermore, in order to sample live animals, the net was left in the water only for 4 hours. Obviously, since this is not a very selective method, other species (both fish and invertebrates) were captured. However, as foreseen by the operational plans of the Antarctic campaign, these samples were used by other researchers who planned to study these animals.

Upon collection, the fish was immediately transported in an aerated bin to the base, where it was placed in a tank supplied with aerated natural seawater at approximately 0°C. After an acclimatization period of seven days, the specimen was anaesthetized and sacrificed as previously reported in the text.

All the required missing information had been added to the materials and methods section.

Section 3.2 - 8 years later 4 adult fish were sampled which were used to confirm target genes. In 8 years several differences might have changed the expression, which is also stated in supplemental data. I believe that this information should also be in the manuscript itself as this is crucial information.

#We accept this criticism. A key sentence in section 2.2 was modified as follows to highlight this potential issue:

“Although the results of this analysis need to be considered with caution due to the different year of sampling compared with the individual subjected for RNA-sequencing, they confirmed the strong tissue-specificity of six selected target genes, which displayed high expression values in all biological replicates, and revealed a limited transcription in the other tissues”.

Line 577: please correct section 2.1 to 3.1

#Fixed

Results:

Line 115: Where is Figure 1B?

#Thank you port pointing this out. Figure 1 was originally planned to have a different structure, being divided in multiple panels, and we forgot to remove the reference to panel B here. This has been fixed.

Line 166-167: Five additional adult individuals? In the methodology , line 575, it was stated that four specimens were collected.

#Thank you. We corrected this mistake.

Supplementary note 2: Please correct RT-PCT into RT-PCR

#Fixed

Line 313: please correct gastrointestinal

#Fixed

Conclusion:

Line 499: please correct preliminary

#Fixed

Line 500: please correct peculiar

#Fixed

Reviewer 2 Report

This is a very interesting study using evolutionary transcriptomics to ascertain the identification and potential role of ice-related markers. The manuscript and the topic are very interesting and fits well to the journal scope. Though difficult to non-RNAseq users the manuscript is easy to follow and understand in most parts.

Comments:

  • I suggest to change the title to really highlight the findings related to the cobalamin and haemoglobin findings.
  • Please, explain why other genes with higher FC in the data are not covered in the study, i.e. CHIT1, FGF18A, PIGB, etc.
  • I do not understand why if the transcobalamin are so highly expressed the conclusion is the lack of cobalamin. Perhaps they produce and have large amounts of cobalamin and need the transporters. This would justify the high load of cobalamin-dependent parasites.

Author Response

Reviewer #2

This is a very interesting study using evolutionary transcriptomics to ascertain the identification and potential role of ice-related markers. The manuscript and the topic are very interesting and fits well to the journal scope. Though difficult to non-RNAseq users the manuscript is easy to follow and understand in most parts.

#We would like to thank the reviewer for his/her positive assessment of our work. We tried to address all the points raised below. All changes are marked in the main text with the “track changes” mode.

Comments:

I suggest to change the title to really highlight the findings related to the cobalamin and haemoglobin findings.

#Thank you for this suggestion. We changed the title as follows:

“Cold adaptation in Antarctic notothenioids: comparative transcriptomics reveals novel insights in the peculiar role of gills and highlights signatures of cobalamin deficiency”

Please, explain why other genes with higher FC in the data are not covered in the study, i.e. CHIT1, FGF18A, PIGB, etc.

#Several out of the 120 DEGs we identified, some of which show high FC values, did not apparently belong to any major regulatory network, hence the interpretation of their up-regulation in Cryonototheniodea was not straightforward. Compared with previous internal drafts of the manuscript, we chose to trim the text, focusing on the description of DEGs that could be either linked with previously described morphological, physiological or metabolic adaptations to cold, or placed in well-defined regulatory networks with interesting implications for Antarctic life. Many other DEGs, not discussed in detail in the present manuscript, are also likely to cover a fundamental role in the context of cod adaptation, but they function in telesost was either unknown (e.g. si:dkeyp-77h1.4) or unclear. For example, we suspected the over expression of CHIT1 to be possibly linked with the high levels of helminth parasitism in Antarctic fish. However, we deemed that discussing this hypothesis in the manuscript, in absence of additional confirmation provided by other genes involved the same immune defense pathways, would have been too speculative.

I do not understand why if the transcobalamin are so highly expressed the conclusion is the lack of cobalamin. Perhaps they produce and have large amounts of cobalamin and need the transporters. This would justify the high load of cobalamin-dependent parasites.

#We updated the text to clarify our interpretation of gene expression data concerning cobalamin metabolism. We believe that the strong upregulation of MTHFSD, BLMH and GMPR clearly points towards a situation of cobalamin deficiency, since BLMH in particular is linked with the detoxification of Hcy-thiolactone, a metabolite typically accumulated in the blood due to folate deficiency.

The massive expression of TCNL in the gills, a tissue where transcobalamins are typically not expressed, may indicate that this tissue is used as an alternative site for cobalamin uptake, that is used instead of the usual uptake site (the digestive tract). We are entirely moving on a speculative level here, but the infestation of the digestive tract by helminth parasites would be expected to nearly completely deprive the fish from the usual route of B12 uptake. Nevertheless, cobalamin is still required for several fundamental physiological processes, and this molecule cannot be synthesized de novo by metazoans.

The gills represent the major surface of contact with the external environment in fish, and they are a relatively parasite-free tissue in Cryonotothenioidea. Therefore, they may provide an optimal alternative site for cobalamin uptake from the external environment, either directly or with the aid of symbiotic bacteria, considering that this molecule is water soluble and potentially bioavailable in the Southern Ocean.

In absence of additional experimental data on this subject, we preferred to avoid discussing this point in detail, but we believe that investigating this topic could be a very interesting aim for future studies.

We modified the final part of section 2.4. as follows, to clarify our hypothesis:
“Under this assumption, the absorption of cobalamin in the gills, or in other tissues, through the interaction between the TCNL-cobalamin complex and a receptor yet to be determined, might mitigate the mitigation of cobalamin deficiency caused by a heavily impaired uptake in other body districts due to widespread dhiplyllobrotriosis. It remains to be investigated whether cobalamin uptake occurs directly from seawater, where this water-soluble molecule is dissolved, or it involves uncharacterized gill-associated cobalamin –producing bacterial symbionts [57].”

Round 2

Reviewer 1 Report

Dear authors

Your manuscript has improved after the modifications, according to reviewers comments. I hereby accept it for publication